# Modulation of the Marine Environment in the Natal Bight

Mark R. Jury [1,2]

1    Physics Department, University of Puerto Rico Mayagüez, Mayaguez, PR 00682, USA; mark.jury@upr.edu
2    Geography Department, University Zululand, KwaDlangezwa 3886, South Africa

**Abstract:** Modulation of the marine environment in the Natal Bight (~29.1°S, 31.6°E) was studied using daily high-resolution climate reanalysis products and monthly satellite green- and red-band reflectance in the period 2002–2022. The KwaZulu-Natal shelf edge is characterized by a narrow band of upwelling next to the warm Agulhas Current. Strong, reversing longshore winds ~7 m/s and meandering poleward flow ~1 m/s pulse the system, but along the leeward coast that forms the Natal Bight, environmental conditions are buffered by a weak cyclonic gyre. Wind and current shear create a shadow zone that aggregates plankton, recycles nutrients, and sustains marine resources. The seasonal cycle is of high amplitude: the surface heat balance reaches $+70\ \text{W/m}^2$ in December, followed by river discharges ~3 M m$^3$/yr of fresh nutrient-rich water that peak in February. This induces a buoyant surface layer that inhibits wind wave turbulence during summer. By contrast, winter (June–August) cooling $-95\ \text{W/m}^2$ and frequent cyclonic storminess deepen the mixed layer from 25 to 65 m, enabling wind wave turbulence to reach the seafloor (Tugela Bank). Red-band reflectance increases 3-fold from summer to winter and is significantly correlated with net heat balance $-0.54$, daily wave heights > 2.5 m $+0.51$, mixed layer depth $+0.47$, sea surface temp $-0.41$, and wind vorticity $-0.39$. Daily longshore winds from the northeast and southwest were, unexpectedly, most amplified in spring (August–October). The seasonality exhibits sequential effects that supports year-round marine nutrification in the Natal Bight. Intra-seasonal fluctuations were related to meandering of the Agulhas Current and changes in longshore winds and shelf waves that impart significant pulsing of near-shore currents at 4–9-day periods. Although the cyclonic gyre in the Natal Bight spins up and down, SST variance was found to be relatively low in its center, where external influences are buffered. Considering linear trends for winds and runoff and surface temperature over the period 1950–2021, we found that northeasterlies increased, runoff decreased, and inshore sea surface temperatures have warmed slowly relative to the adjacent land surface temperature. New insights derive from the use of monthly satellite red-band reflectance and daily 10 km climate reanalysis fields to understand how air–land–sea fluxes modulate the marine environment in the Natal Bight.

**Keywords:** Natal Bight; environmental modulation; marine nutrification



## 1. Introduction

The warm Agulhas Current flows along the southeast coast of Africa as part of the semi-permanent anticyclonic gyre in the SW Indian Ocean. At Cape St. Lucia, 28.5°S and 32.3°E, the coast bends southwest and the shelf widens ~40 km creating the Natal Bight (Figure 1a) where currents are weak and biomass accumulates [1]. A semi-permanent cold eddy is formed by cyclonic wind and current shear [2–4] that adjusts to the meandering of the Agulhas Current [5–7]. The Tugela River discharges at 29.2°S and its sediment-laden plume tends to drift northward along the coast despite alternating longshore winds [8–11] that modulate Ekman transport and sea surface temperature (SST) [12,13].

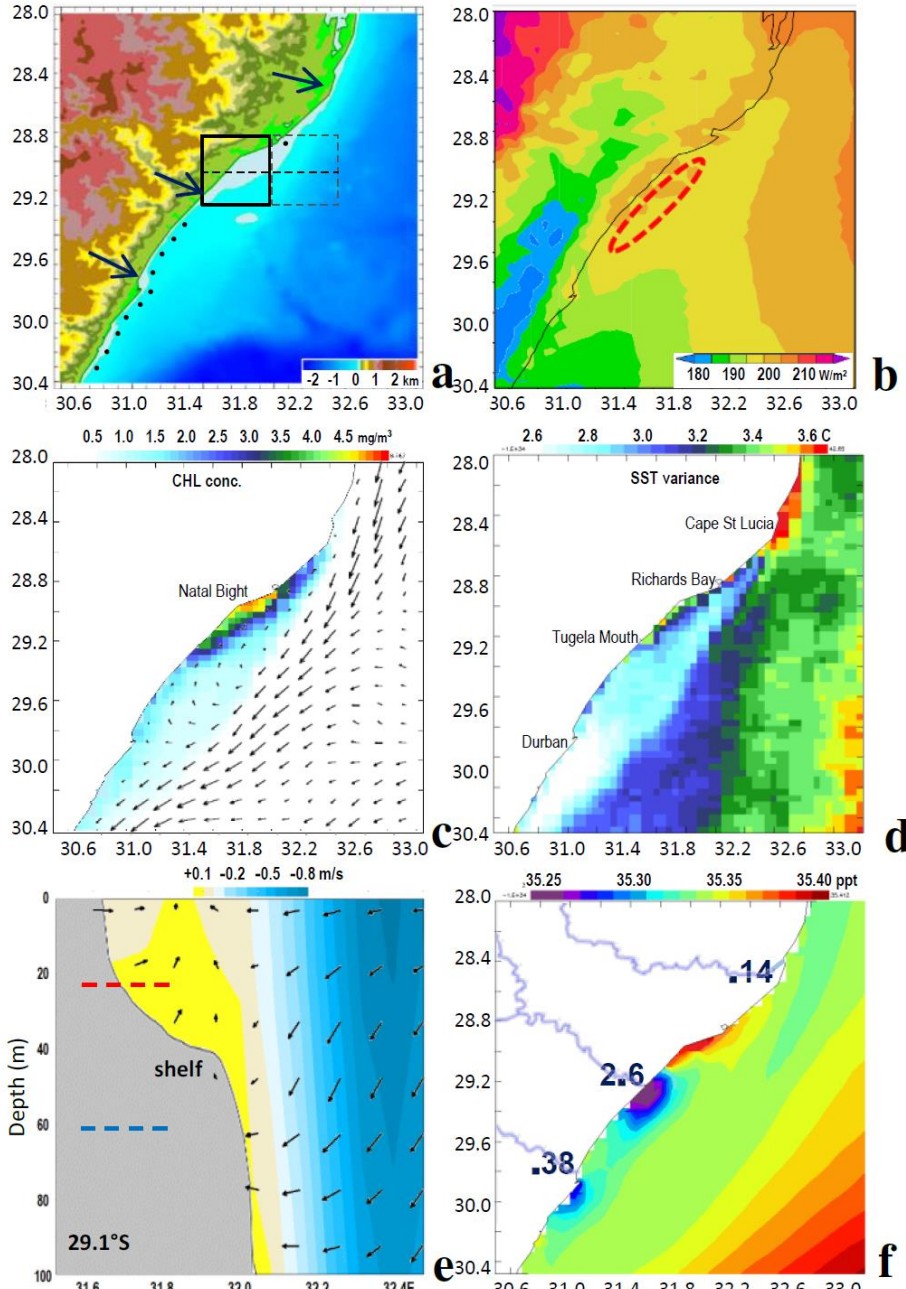

**Figure 1.** (**a**) Elevation map of study area, shelf < 60 m: greyscale; 'Natal Bight': solid box; offshore: dashed box; vertical section: dashed; main rivers: arrows; effluents: dots. The 2002–2021 mean: (**b**) net solar radiation (W/m$^2$) and MLD > 40 m (dashed zone); (**c**) MODIS estimated CHL concentration (mg/m$^3$) and HYCOM currents (largest 1 m/s); (**d**) SST variance and place names; (**e**) depth section on 29.1°S of HYCOM meridional current (shaded m/s) and zonal circulation (vectors, largest 0.2 m/s) with bathymetry, MLD dashed red summer/blue winter; and (**f**) HYCOM near-surface salinity (ppt) with major rivers and discharge labels (M m$^3$/yr).

The synthesis by [14] identified environmental processes that affect biomass nutrification in the Natal Bight. Coastal upwelling at Cape St. Lucia is induced by intermittent northeasterly winds and steady offshore shear in the Agulhas Current [11,15]. The 40 km coastal recess comprising the Natal Bight retains a leeward cyclonic gyre about half the time [4]. It generates northward flow near the coast which is amplified by oblique swells. Although the Agulhas Current carries seawater that is low in chlorophyll/phytoplankton, the inshore gyre recirculates nutrient-rich sediment from coastal runoff (~4 B m$^3$/yr) and

upstream upwelling [16]. Its cyclonic spin lifts seawater over the Tugela Bank throughout the year [17]. Time series from moorings and coastal stations show that most upwelling events (SST < 20 °C) relate to fair-weather shallow northeasterly winds that increase seaward-producing offshore Ekman transport.

Geological influences on Natal Bight oceanography are unique. Relic fluvial sediments, mainly from the Tugela River, form a broad shelf that is available for reworking during winter when a negative heat balance deepens the mixed layer. Huge ocean swells from passing winter storms generate northward surf zone currents > 1 m/s [18] and turbulence on the seabed, which scours the fine sands, mud, and detritus. Sediment size distributions can be traced to storm tides, reworking of submerged relic shorelines, and shelf edge transport by the Agulhas Current. The inshore cyclonic gyre retains fine material introduced by summertime flood events, where multiple rivers can discharge >1000 m$^3$/s over many days. The buoyant river plumes tend to be red-tinted due to upstream mineral deposits.

Biological surveys in the Natal Bight reveal high abundance and diversity [19,20]. Plankton tend to concentrate along the coast near the Tugela Mouth and in the shelf edge upwelling plume advected from Cape St. Lucia. By contrast, the center of the cyclonic gyre is less productive. Commercial fisheries data have been used as tracers of oceanographic influences over the shelf [14]. Species composition relates to distance from the Tugela Mouth and species richness increases with depth consistent with seasonal nutrient levels [21,22]. Nitrate levels increase during winter and inshore near Richards Bay due to upwelling [15]. Other nutrients tend to peak during summer near river mouths [23,24]. Nutrient uptake by plankton in the Natal Bight was determined to be efficient, except under turbid conditions [25,26]. Phytoplankton accumulate during winter in the mixed layer north of the Tugela Mouth [21,22,25,26]. Phytoplankton concentrations were found to depend on sea temperature, net solar radiation, and nutrients, while zooplankton exhibited adaptability to fast changing weather [25] and had greater biomass in winter due to marine organic carbon and nitrogen [24].

The year-round nature of nutrient recycling in the Natal Bight sustains primary production through periods of low runoff and weak upwelling [27]. Similarly, demersal fisheries benefit from recirculated estuarine inputs despite offshore export and show low levels of predator–prey isotope enrichment. Ref. [28] investigated the mineral content of sediment and nutrient uptake by marine species and found higher values near the Tugela Mouth. An ecosystem model predicted a large benthic biomass supported by runoff and recycling, typical of subtropical coastal bays with cyclonic currents and upstream upwelling. Our understanding of Natal Bight oceanography has matured, yet questions remain on how nutrients are sourced and entrained in the euphotic zone [14].

The ~ZAR50 B economic output of KZN province is derived mainly from the coastal zone and its natural resources [24,29–31]. Environmental modulation of KZN fisheries depends on upstream sources and dispersion by the narrow Agulhas Current [32–34]. Institutional monthly fish catch data (mainly Shad) indicate a winter peak (June–August) culminating in the 'sardine run' [35–38]. The study by [39] found that greater KZN fish catch corresponded with warmer saltier waters. However, that study used low-resolution data and 5 × 5° indices representing the offshore environment.

High-resolution coupled reanalysis products have recently become available to characterize the shelf environment [11,40]. Operational data assimilation of satellite reflectance blended with in situ observations of momentum, heat, and water fluxes over a rolling multi-day window, offer a variety of gridded field data near the coast [41,42]. Here we investigate environmental controls on marine nutrification in the Natal Bight using those products.

## 2. Data and Methods

The Natal Bight environment (28–30.5°S and 30.5–33°E) is analyzed at a resolution of ~10 km horizonal, 10 m vertical, daily, using the Hybrid-coordinate ocean model v3 (HYCOM) underpinned by coupled data assimilation [43–45]. HYCOM provides fields of

sea temperature, salinity, sea surface height, currents, vertical motion, mixed layer depth (MLD) [46], and wind stress. Near-surface currents (layer averaged 1–20 m depth) are derived via model physics that blend dynamical forcing from satellite altimetry, in situ measurements reported to SADCO, Ekman-response to scatterometer-derived wind stress, and constraints of static bathymetry.

NASA level-3 MODIS and VIIRS satellite color reflectance and infrared SST data [47] at 4 km resolution provide a statistical basis for this study. Time series of green-band reflectance was formed by averaging chlorophyll (443 nm) and fluorescent line-height (CHL+FLH) [48–50], while red-band reflectance was derived by averaging 667–671 nm radiance from MODIS and VIIRS to quantify nutrification by suspended sediments [51,52]. Inspection of image sequences found that weekly composites were degraded by clouds, so monthly values were obtained in the period 2002–2021. Time series were generated by averaging over the Natal Bight: 29.3–28.8°S and 31.4–31.9°E (Figure 1a): a 50 × 50 km index area with ~1/3 land 2/3 sea, stretching from the Tugela Mouth to Richards Bay.

Atmospheric data derive from the Coupled Forecast System (CFS2) [53] and European Community reanalysis (ERA5) [54] at a resolution of ~25 km and 10–250 m vertical daily, while hydrology data come from the FEWS land data assimilation system (FLDAS) [55] and from [56]. The variables include sea level pressure, air temperature, humidity, evaporation, precipitation, surface wind velocity, vertical motion, net heat balance, net solar radiation, boundary layer height, runoff, and river discharge. Wave action is characterized by Wavewatch 3 (W3) reanalysis [57] which assimilates buoy and ship observations, satellite microwave measurements and weather data in a multi-spectral model [58,59]. Parameters include significant wave height, wave period, and wave direction at 25 km daily resolution, that compare favorably with wave rider observations [60]. Coverage by polar-orbiting satellite radiometers has improved since the 1990s, lending confidence to daily retrievals of near-shore currents, wind waves, salinity, etc.

River discharge time series were formulated by averaging the Natal Bight and Tugela Mouth. For winds and currents, data from the adjacent offshore area were extracted to calculate $\partial V/\partial x$ vorticity (shear of longshore flow, where <0 is cyclonic). The ERA5, FLDAS, and HYCOM time series were verified against alternative products having inshore grid points: CFS2, CHIRPS2, GODAS, GPM, MERRA2, ORA5, and SODA3 as outlined in [11]. Most parameters demonstrated coherence except salinity and MLD, wherein median values from multiple reanalysis [61] were calculated.

In addition to monthly data, daily time series of surface winds, near-surface currents, and wave height in the Natal Bight were obtained and subjected to Fourier analysis to determine spectral power from 1 to 30 days. For fluctuating longshore winds and wave heights, monthly indices were generated from the sum of daily exceedances in the Natal Bight: V wind < −5 and >+5 m/s, and wave height > 2.5 m. Daily data were summarized into histograms and scatterplots to illustrate distributions, threshold effects and cross-correlations.

Reanalysis and satellite data were used to calculate mean fields 28–30.5°S and 30.5–33°E and east–west depth sections on 29.1°S (cf. Figure 1a). Hovmoller plots of monthly MODIS (VIIRS) red-band reflectance in the first (second) half of record were compared with river discharge. Pair-wise cross-correlations were calculated for Natal Bight monthly time series (cf. Supplementary Material) to establish the degree of influence. Our record length of 240 months has ~40 degrees of freedom that requires a correlation coefficient r >| 0.30 | for significance at 95% confidence. Annual cycles were calculated over 20 years and statistical variance (average squared deviations from mean) was mapped for MODIS SST and for other variables along the 29.1°S slice. Acronyms and information on the data are listed in Table 1.

**Table 1.** Acronyms and dataset details; some are reduced to monthly.

| Acronym | Name and Variables | Resolution |
|:---:|:---:|:---:|
| CFS2 | Coupled forecast system reanalysis v2<br>Wind, current, salinity, MLD | 30 km<br>daily |
| ECMWF5 | European Community reanalysis v5<br>Temp, wind, heat flux, discharge, etc. | 25 km<br>daily |
| FLDAS | FEWS Land Data Assimilation System<br>Rainfall, runoff | 10 km<br>daily |
| HYCOM3 | Hybrid coupled ocean model v3<br>Sea temp, current, salinity, MLD, etc. | 10 km<br>daily |
| MODIS | Moderate Imaging Sensor (satellite)<br>ocean color: green- and red-band 667 nm | 4 km<br>weekly |
| NASA | National Aeron. Space Admin.<br>GPM rain, heat flux | 10 km<br>daily |
| NOAA | National Oceanic and Atmospheric Admin.<br>satellite SST, net OLR, GODAS rean. | 25 km<br>weekly |
| SADCO | South African Data Centre for Oceanography | station, ship |
| SODA3 | Simple Ocean Data Assimilation v3<br>Current, vertical motion, salinity, MLD | 25 km<br>monthly |
| VIIRS | Visible Infrared Imaging Radiometer Suite<br>satellite ocean color: red-band 671 nm | 4 km<br>weekly |
| W3 | Wavewatch ocean reanalysis v3<br>wave height, period, direction | 25 km<br>daily |

The process by which seafloor nutrients are lifted into the euphotic zone depends on wind waves, thermohaline stability, and (inefficient) momentum transfer [62]. To study these effects, mean vertical profiles of sea temperature and V currents in the Natal Bight were made for summer and winter 2002–2021, and seasonal histograms of daily wave direction were analyzed.

Inter-annual variability was quantified by Empirical Orthogonal Function (EOF) analysis applied to monthly standardized SST and U V wind fields, 28–30.5°S and 30.5–33°E. The second mode of SST (dipole) and first mode of U V winds (northeasterly) were extracted to determine coherence. Wavelet spectral analysis was applied to the time scores to determine multi-year oscillations. The winter red-band reflectance time series was regressed onto fields of ERA5 precipitation, net heat balance, and sea level air pressure using June–August values from 2002–2021. For rainfall the preceding December–February summer was considered.

A dry to wet climate transition was analyzed for March–April 2022 via monthly patterns and daily fluctuations. Runoff anomalies were calculated, and maps of recirculating currents were obtained for 14 March and 23 April 2022. Depth (or height) sections were analyzed on 29.1°S of variance in HYCOM sea temperature, salinity, U currents, and ERA5 V winds.

Exploratory work found that cyclonic wind vorticity ($-\partial V/\partial x$) prevails over the Natal Bight. To better understand its seasonal cycle, mean U winds for summer and winter were calculated on 29.1°S and the mean diurnal cycle was analyzed. Daily wind vorticity values over the Natal Bight were calculated and weather maps were inspected for cyclonic events. Lastly, long-term trends were mapped for surface temperature, runoff, and marine winds from 1950–2021 by linear regression to understand the local uptake of global warming.

## 3. Results

### 3.1. Climatology and Annual Cycle

Figure 1a presents the study area, with steep topography in the west and steep bathymetry in the northeast. The south-facing Natal Bight and its wider shelf is our focus for spatio-temporal analysis. Mean maps 2002–2021 are presented in Figure 1b–f. The net solar radiation exceeds 200 W/m$^2$ over the Natal Bight and the mixed layer varies from 25 m in summer to 60 m in winter. Orographic clouds and humidity fluxes create a longshore gradient of net solar radiation from 185 (southwest) to 205 W/m$^2$ (northeast). Currents in the Natal Bight are weak and accumulate phytoplankton (CHL > 3 mg/m$^3$), indicative of a leeward shadow zone from the narrow fast-flowing Agulhas Current. Mean SST variance, based on weekly MODIS imagery 2002–2021, is low in the shadow zone that extends to Durban. SST variance is much higher along the north coast due to upwelling, and outside the Agulhas due to meandering. The depth section reveals sharp gradients in the meridional current and shoreward downwelling at the shelf edge. Inshore currents are weak northward and upwelling. Over the Tugela Bank, the zonal circulation is convergent, favoring biological accumulation. River discharges (~3 M m$^3$/yr) induce freshwater lenses off the Tugela Mouth and Durban, whilst the Agulhas advects values ~35.32 ppt keeping saltier waters offshore. An interesting feature captured by HYCOM reanalysis is a salty lens on the south-facing coast from Richards Bay to Mtunzini. In overview, the high-resolution climatology maps indicate that multiple environmental factors diffuse surface nutrients: vigorous runoff into a deep mixed layer and nearshore upwelling within cyclonic recirculating currents with little variance. The mean CHL map defines the Natal Bight as the south-facing part of the coast from Tugela Mouth to Richards Bay, there is no evidence of a 'southern' Natal Bight.

In Figure 2a,b red-band reflectance is analyzed in context of river discharge. The Hovmoller plot on 29.1°S in the VIIRS era 2012–2022 reflects a seaward spread of ~50 km from the shore. There are low values (<2 × 10$^{-4}$ sr$^{-1}$) nearshore during early summer November–December 2013, 2016, 2018–2021. High values (>20 × 10$^{-4}$ sr$^{-1}$) appear almost every winter June–August 2014, 2020, 2022; cropped at ~32.1°E by the Agulhas Current. A typical sequence of red-band reflectance maps reveals high values north of the Tugela Mouth in late summer. Cyclonic currents in the Natal Bight sweep the inorganic sediments and biomass into shelf edge plumes that extend past Durban (30°S). These consolidate during winter into a half-moon shape that is ~100 km longshore and ~50 km seaward, similar to the mean green-band reflectance. Aerial photos of the Tugela Mouth (Figure 2c) indicate the river discharge as the main source of red-band reflectance, while the Hovmoller plot suggests a 6-month lag between peak summer runoff and winter nutrification. This is quantified via mean annual cycles in Figure 3a–h.

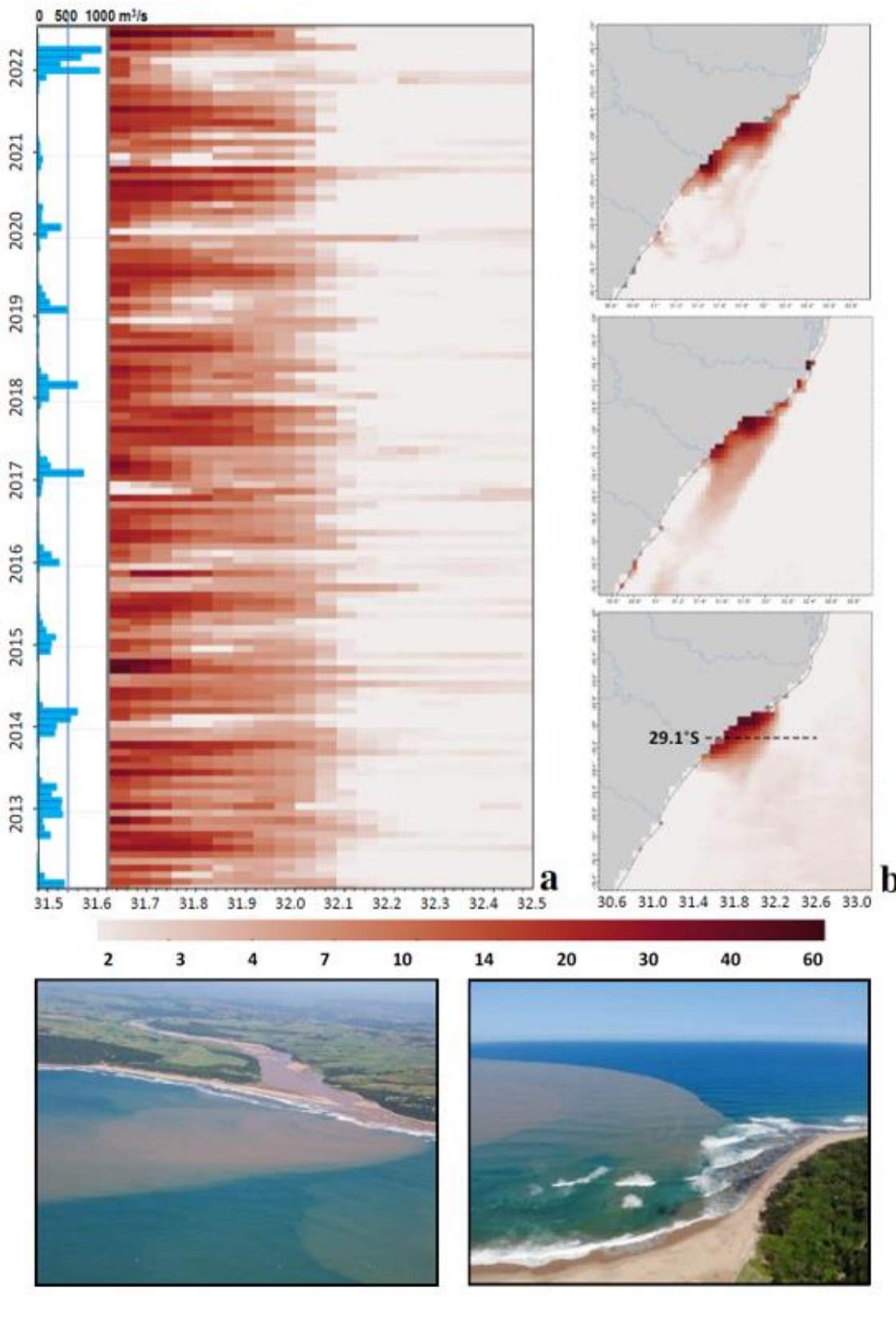

**Figure 2.** (**a**) Hovmoller plot of monthly VIIRS satellite red-band reflectance (log sr$^{-1}$ × 10$^{-4}$) on 29.1°S, with Tugela discharge on left. (**b**) Sequence of red-band reflectance (top-down) February, April, June 2013. (**c**) Aerial photos of the Tugela sediment plume.

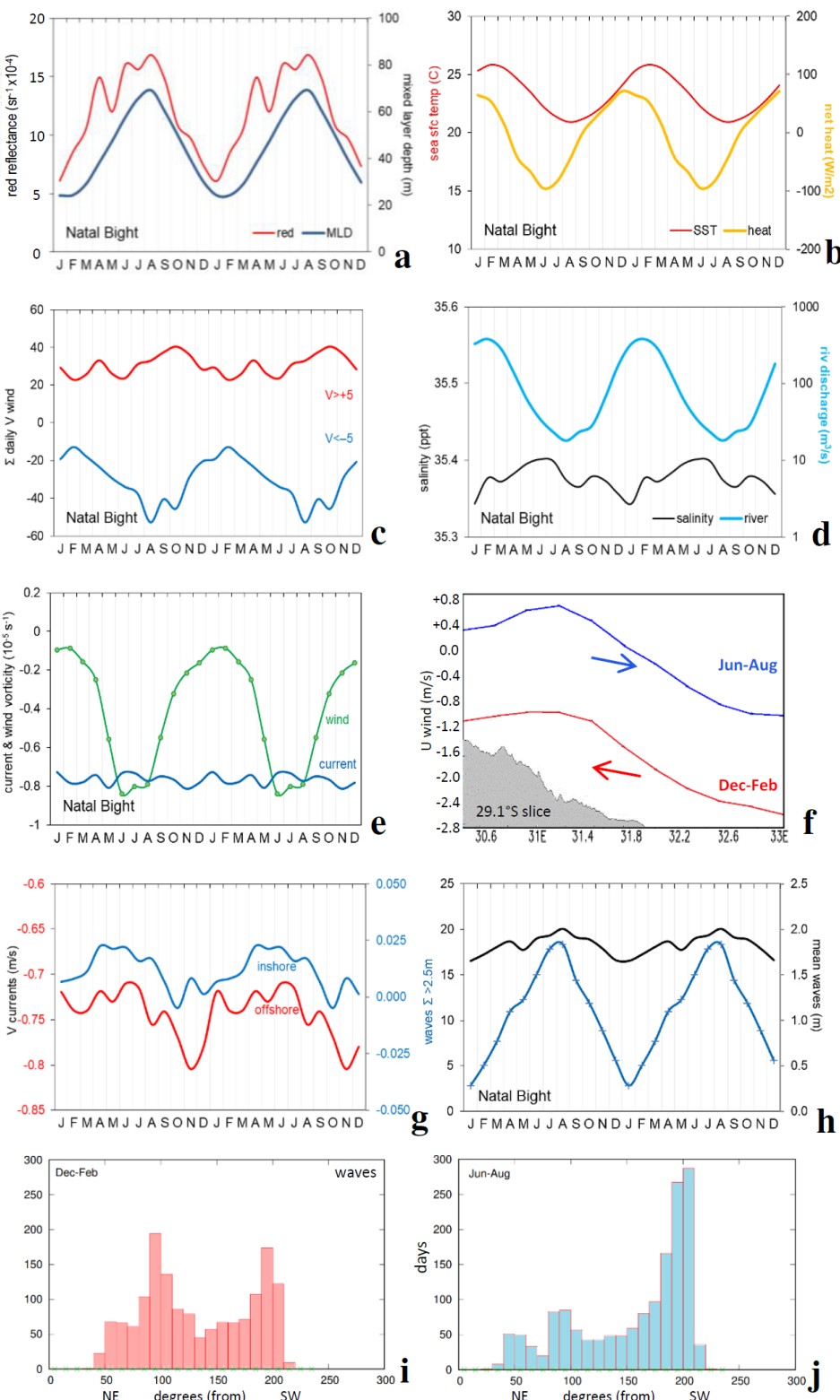

**Figure 3.** Mean annual cycles in the Natal Bight of: (**a**) red-band reflectance/mixed layer depth, (**b**) SST/net heat balance, (**c**) sum of daily V wind < −5 > +5 m/s, (**d**) mean river discharge and salinity, (**e**) cyclonic vorticity of wind and current, (**f**) mean zonal wind on 29.1°S in summer and winter with topographic profile, (**g**) inshore and offshore V current, and (**h**) mean wave height and sum of daily values >2.5 m, differing scales. (**i**,**j**) Daily wave direction histograms for summer and winter in the Natal Bight, 2002–2021.

The red-band reflectance shows a minimum in December–January, a minor peak in April and a crest from June–August (Figure 3a). Many environmental variables exhibit a large seasonal cycle which underpins marine nutrification. There is a heat deficit $<-90$ W/m$^2$ in June–July which cools SST $<$ 22 °C in August–September (Figure 3b). MLD and wave height $>2.5$ m peak in July–August (Figure 3a,h), stirring up sediments near the coast. River discharge and SST crest in late summer when winds tend to be light (Figure 3b,d). Yet Natal Bight salinity seems decoupled from river discharge, possibly due to the salty lens associated with coastal upwelling (cf. Figure 1f) that keeps values in a narrow range. Meridional winds (above daily threshold) reach maximum northward and southward values in August–October (Figure 3c).

Current shear is persistently cyclonic $-0.8\ 10^{-5}$ s$^{-1}$ as expected; however, the mean annual cycle of wind vorticity in the Natal Bight alternates from ~0 in summer to $-0.8\ 10^{-5}$ s$^{-1}$ in winter (Figure 3e). Mean zonal winds along the 29.1°S slice (Figure 3f) reveal why. During winter, nocturnal land breezes (+U) drain from the cool interior and push longshore winds seaward, ensuring larger vorticity. During summer diurnal sea breezes (−U) are drawn toward the warm interior and pull the longshore winds coastward, weakening vorticity over the Natal Bight. Currents exhibit a weak annual cycle: northward in early winter/southward in early summer (Figure 3g) amidst sustained shear $\partial V/\partial x$. Even daily currents stay in a narrow range, according to HYCOM reanalysis.

Daily wave direction histograms verify large seasonal differences (Figure 3i,j); being equally from the east and south in summer but distributed $\pm190°$ in winter, inducing northward drift along the coast. The spectral and histogram analysis of daily time series 2002–2021 for Natal Bight meridional winds, currents, and wave heights (Figure 4a,b) quantify pulsing of the environment. V winds show significant 4–9-day pulsing and a broad distribution of longshore winds, with nearly 1/4 above $\pm5$ m/s threshold. The Natal Bight is more exposed to +V than −V due to horizontal shear that accompanies airflow around the Mascarene high (e.g., northeasters are weak inshore especially during winter land breezes). In contrast, southwesterly winds often arrive as a narrow jet known as a 'buster' [63]. V currents exhibit significant 5–7-day pulsing which derives from wind-forcing and meandering by the offshore current. The HYCOM current histogram is relatively gaussian and tending to northward but weak, suggesting little export from the Natal Bight. Wave heights have low spectral energy, from a mixture of local and remote storms that may be out-of-phase. There is significant 5-day pulsing and 20–30-day variability. Wave height histograms reflect energetic conditions averaging ~1.8 m and many days $>2.5$ m offering turbulent mixing.

Three-way scatterplots identify the conditional relationship between daily winds, currents, and evaporation in the Natal Bight (Figure 4c). The r$^2$ fit achieves 0.418, so inshore currents may be defined as wind-driven and most northward when latent heat flux is greatest (e.g., turbulent dry airflow over warm water leading to cooling). The transfer of momentum from air to sea is more efficient under breaking waves during winter storms (whitecaps > 7 m/s). The mean vertical profile of V currents (Figure 4d) exhibit departures from near-zero value in the upper 15 m. Below the mixed layer there is mean northward flow (+0.05 m/s) that tends to deepen in winter. The mean vertical profile of sea temperature in the Natal Bight (Figure 4d) is isothermal in winter, but stably stratified in summer (2 °C/35 m). Thus air–sea momentum transfer is inhibited during summer and is, in any case, diminished by the annual cycle of waves (cf. Figure 3f). One outcome of ~5-day pulsing is that multi-day ship surveys may not be synoptic, e.g., weather changes and current meanderings tend to be aliased.

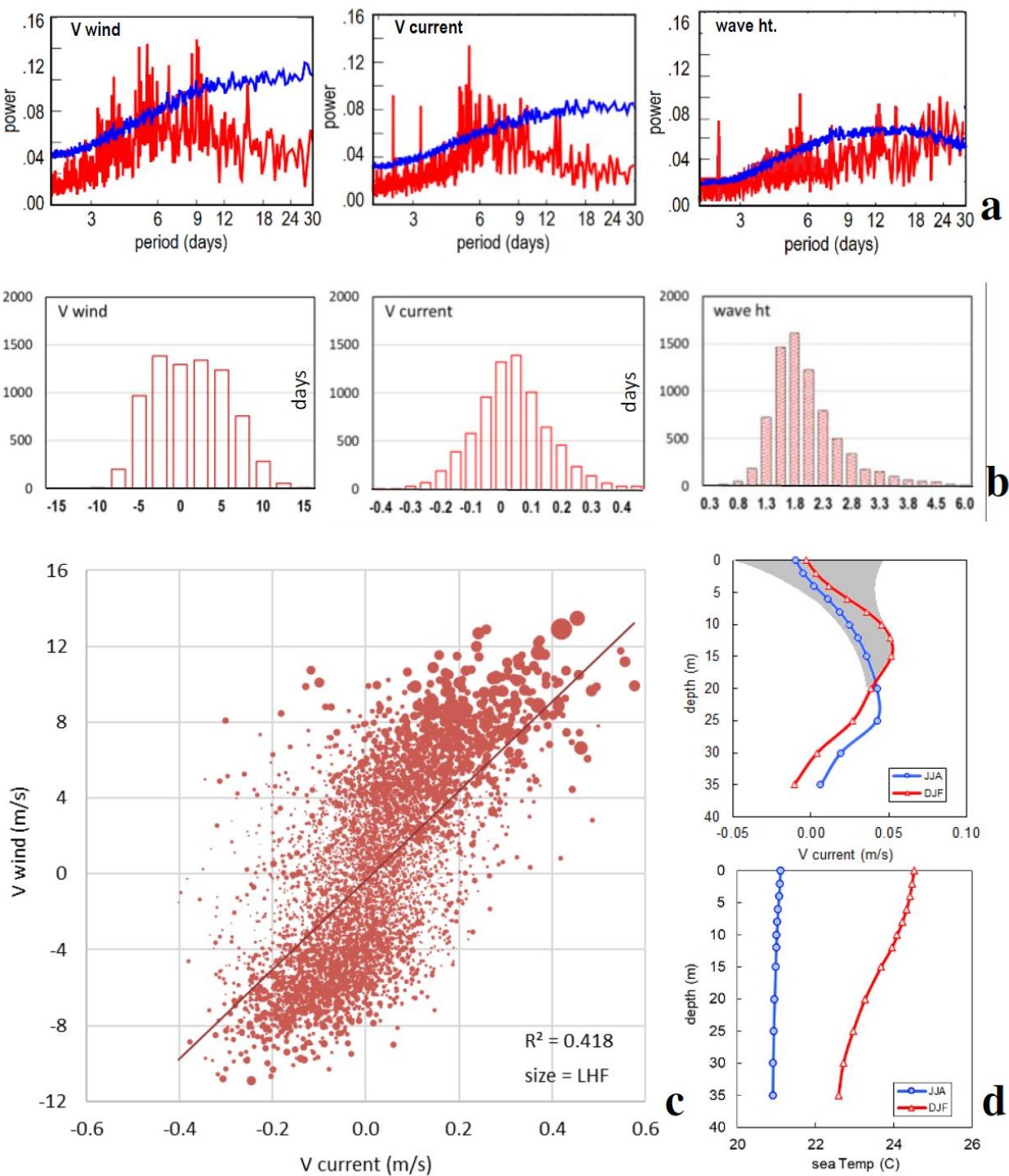

**Figure 4.** Analysis of daily time series 2002–2021 in the Natal Bight; spectral power at periods from 1 to 30 days with blue 95% confidence for: (**a**) V wind, V current, and wave height (left to right). Histograms for (**b**) V wind -m/s, V current -m/s, and wave height -m (left to right). (**c**) Three-way scatterplot of daily V current (−10 m) and V wind (+10 m) sized by latent heat flux (0 m) with linear fit, from HYCOM, ERA5, and CFS2, respectively. (**d**) Seasonal mean profiles of V current and sea temp in the Natal Bight with shaded current variance (in response to fluctuating longshore winds) at virtual station.

### 3.2. Cross-Correlations and Inter-Annual Variability

Monthly time series of color reflectance, SST, and environmental variables are illustrated in Supplementary Information Figure S1 and pair-wise correlations are listed in

Table 2. Although many of the statistical outcomes depend on seasonality, a key point is that green-band reflectance (CHL+FLH) does not relate significantly with any variable unlike results for Cape St. Francis (Jury 2019a) whose coastal upwelling supports a pelagic fishery. In contrast, red-band reflectance in the Natal Bight shows significant correlations with net heat of −0.54, wave height of +0.51, MLD of +0.47, SST of −0.41, and wind vorticity of −0.39. SST also exhibits strong relationships with MLD, river discharge, V wind, and vorticity. So why does green-band reflectance not correlate with the physical environment? Considering its temporal nature (S1), we note CHL+FLH fluctuations have an ill-defined seasonality. Perhaps red-band reflectance represents sediment nutrification that contributes to productivity in the Natal Bight.

**Table 2.** Simultaneous cross-correlation of monthly color reflectance, SST, and environmental variables in the Natal Bight. Bold values are significant at 95% confidence, 2002–2021. Most time series are illustrated in Supplementary Materials.

| Parameter | Green | Red | SST |
|:---:|:---:|:---:|:---:|
| Red-reflectance | **0.36** | | |
| SST | −0.18 | **−0.41** | |
| Salinity | 0.07 | 0.04 | −0.09 |
| M.L.Depth | 0.17 | **0.47** | **−0.84** |
| River disch. | −0.17 | −0.27 | **0.65** |
| Precipitation | −0.11 | **−0.30** | **0.47** |
| Wind-vorticity | −0.13 | **−0.39** | **0.70** |
| Net heat | −0.14 | **−0.54** | **0.46** |
| Vwind > 5 | 0.05 | 0.22 | −0.22 |
| Vwind < −5 | −0.21 | −0.24 | **0.61** |
| $V_i$ current | −0.03 | 0.18 | 0.04 |
| Wave > 2.5 | 0.19 | **0.51** | **−0.43** |

Interannual variability is studied via EOF analysis of SST and wind, and by field correlations with winter red-band time series. SST PC2 representing an inshore/offshore dipole and wind PC1 representing longshore wind shear are analyzed in Figure 5a–d. The standardized time scores correspond well despite differences in explained variance (SST PC2 11% vs. wind PC1 84%). Both exhibit annual cycling that peaks in winter. SST PC2 has significant spectral energy at 3.5–5 yr, while wind PC1 is 2–3 yr. The cool inshore/warm offshore pattern is associated with shallow northeasterly winds and cyclonic vorticity under high pressure-induced airflow. The SST dipole would accelerate the Agulhas Current and spin up the cyclonic gyre, lifting nutrients into the euphotic zone. Whilst negative phases of wind lead to opposing responses, the poleward shear-edge current generates steady cyclonic shear.

Field correlations with June–August red-band time series (Figure 6a–c) define interannual forcing. The preceding December–February rainfall shows a marked wet tongue over the Tugela Valley, thus confirming that runoff leads nutrients and that ERA5 reanalysis correctly places the signal at the Tugela Mouth. Other field correlations are calculated simultaneous (e.g., June–August) and show a net heat balance dipole (deficit inshore/surplus offshore) and high pressure ridging in the southwest, contributing to increased red-band reflectance during winter. We interpret that inshore cooling deepens the mixed layer and the ridging high delivers large waves behind a frontal trough to stir up sediments over the Tugela Bank.

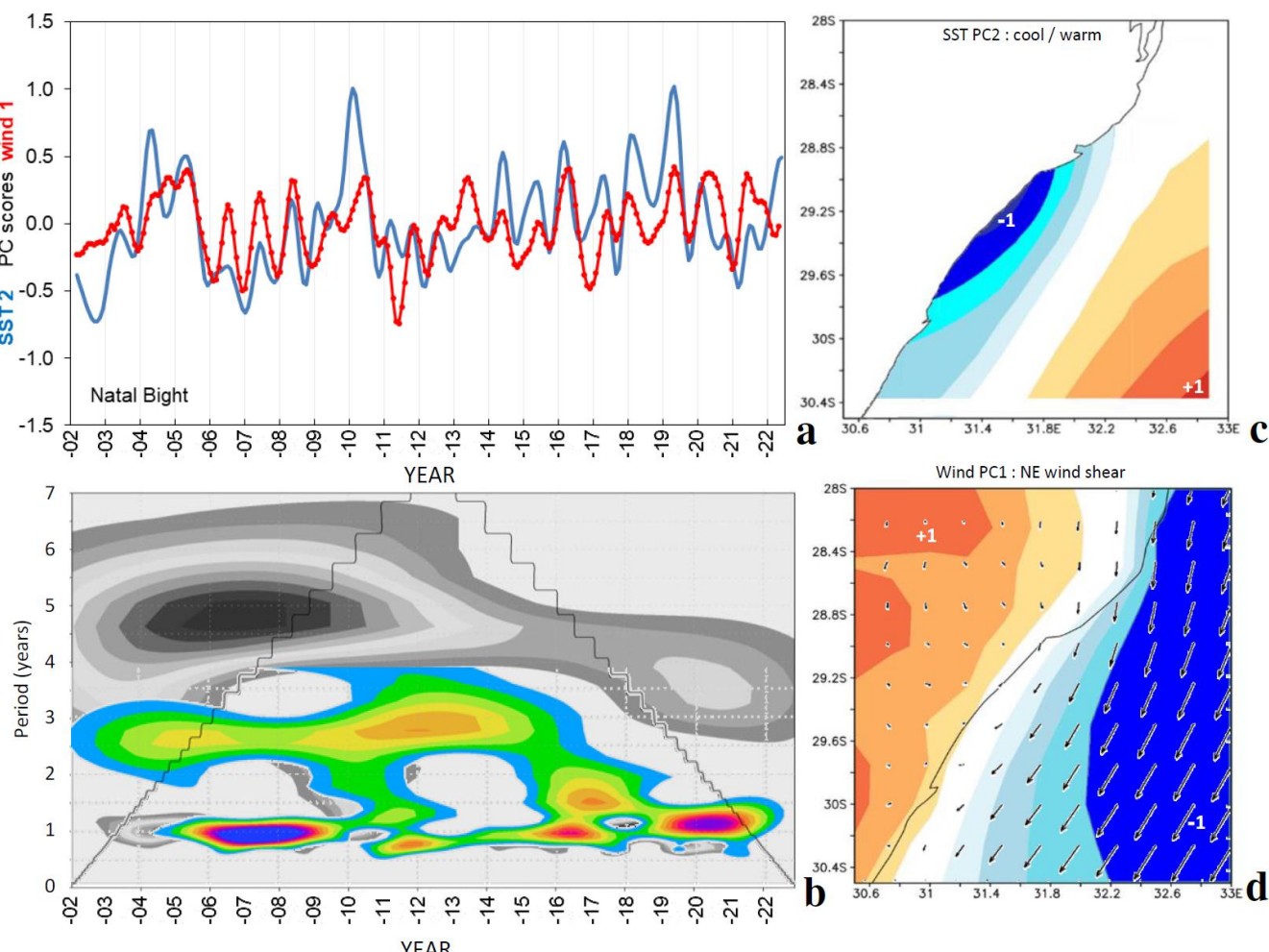

**Figure 5.** (**a**) Principal component time scores for standardized SST PC2 and wind PC1, and (**b**) wavelet spectral power shaded >95% confidence. Grey: SST; color: wind. Principal component spatial loading patterns for (**c**) SST PC2, and (**d**) wind PC1 (meridional shaded, total vector largest 7 m/s).

### 3.3. Sharp Weather Transition: March–April 2022

The April peak of red-band reflectance (cf. Figure 3a) may be related to the overlap of summer runoff and winter mixing. In March–April 2022 the Natal Bight experienced a quiescent spell followed by stormy weather, offering a case study in sudden environmental changes. Figure 7a–d illustrates sharp differences in red-band reflectance, SST, winds, and rainfall from one month to the next. March 2022 had dry northeasterly winds and sunny skies. Red-band reflectance $>10 \times 10^{-4}$ sr$^{-1}$ was confined north of the Tugela Mouth, and the offshore current had SST ~27 °C. In sharp contrast, April 2022 had southerly winds and cloudy skies with rainfall ~500 mm, causing destruction in Durban. Offshore SST cooled by 1.5 °C; the daily time series, Figure 8a–d, tell the story.

Tugela discharge was minimal in March but surged to 1500 m$^3$/s in April 2022, pulsing four times at ~6-day intervals, consistent with a 'family' of subtropical low-pressure cells passing eastward. Meridional winds oscillated from north to south and reached gale force a few times, accompanied by 3+ m waves that can stir sediments to ~30 m depth. Northward currents exceeded 0.2 m/s about 20 km off the coast (Figure 9a,b). Although winds and waves are relevant, the axis/shape/speed of the Agulhas Current played a role in sustaining the cyclonic gyre in the Natal Bight on 14 March and 23 April 2022. The offshore current was narrow and made a gradual cyclonic curve that trapped the inshore gyre. Runoff anomalies show vivid contrasts in the weather from March to April 2022.

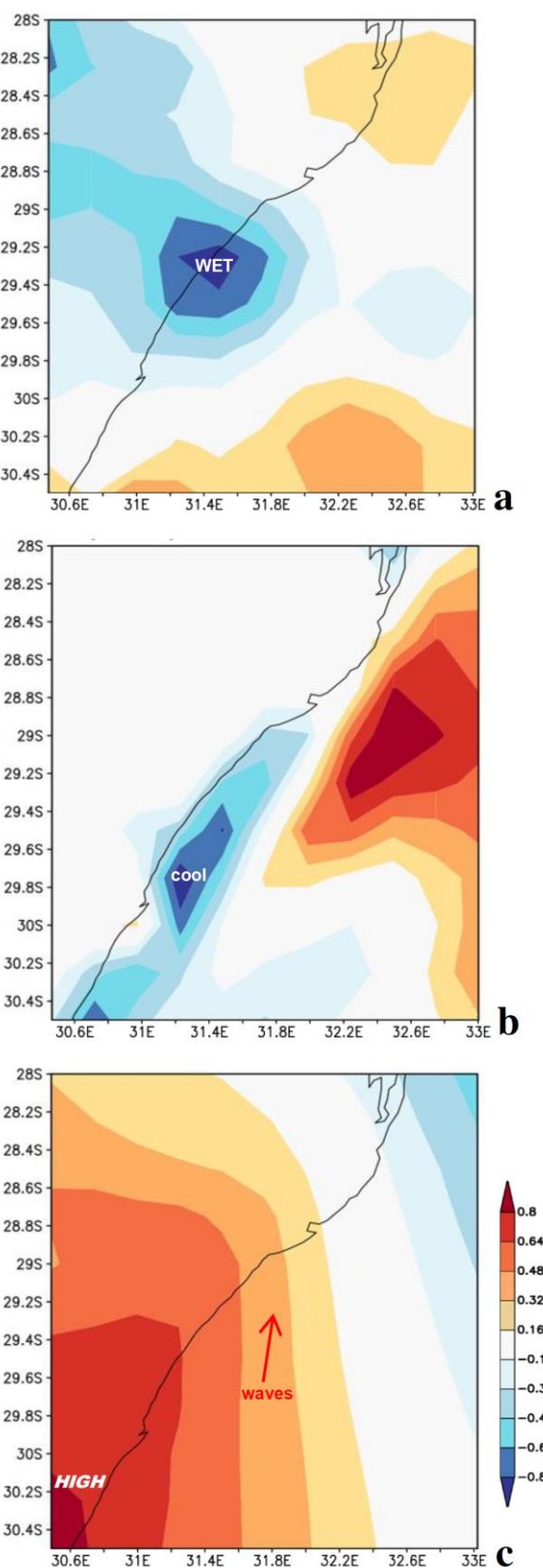

**Figure 6.** Correlation of winter red-band reflectance June–August 2002–2021 onto ERA5 fields of: (**a**) preceding summer rainfall (color-bar reversed), (**b**) winter net heat balance, and (**c**) air pressure. Lower color bar refers to all, units are correlation fraction.

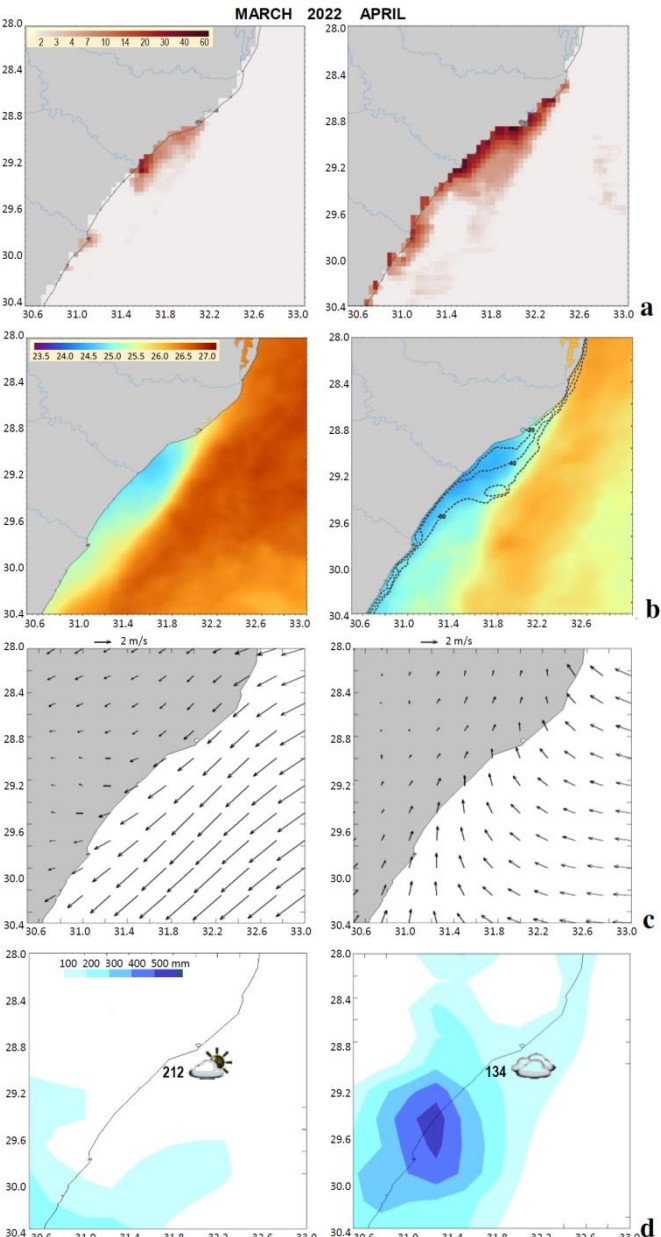

**Figure 7.** Case study of March (left) and April (right) 2022: (**a**) monthly VIIRS red-band reflectance (log sr$^{-1}$ $\times$ 10$^{-4}$), (**b**) GHR SST with shelf bathymetry, (**c**) ERA5 surface wind vector, (**d**) cumulative rainfall (shaded mm) with number = net solar radiation in the Natal Bight (W/m$^2$), indicating weather change from dry (March) to wet (April).

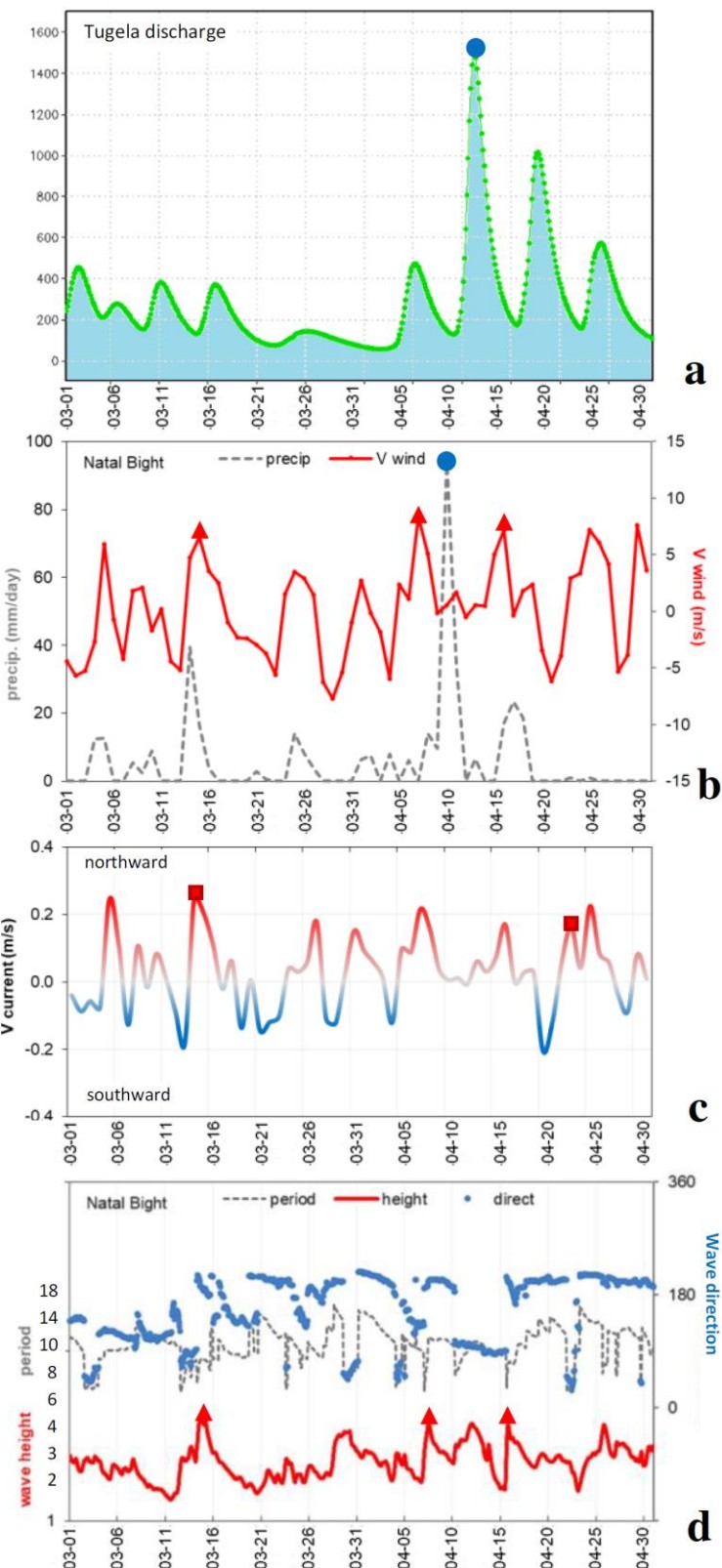

**Figure 8.** Case study daily time series March–April 2022, (**a**) Tugela River discharge (m$^3$/s); Natal Bight area: (**b**) ERA5 V wind and precipitation, (**c**) HYCOM V currents, (**d**) W3 wave characteristics. Wind/wave events denoted by ▲ in (**b**,**d**), flood event by ● in (**a**,**b**) northward currents featured in Figure 9 by ■ in (**c**).

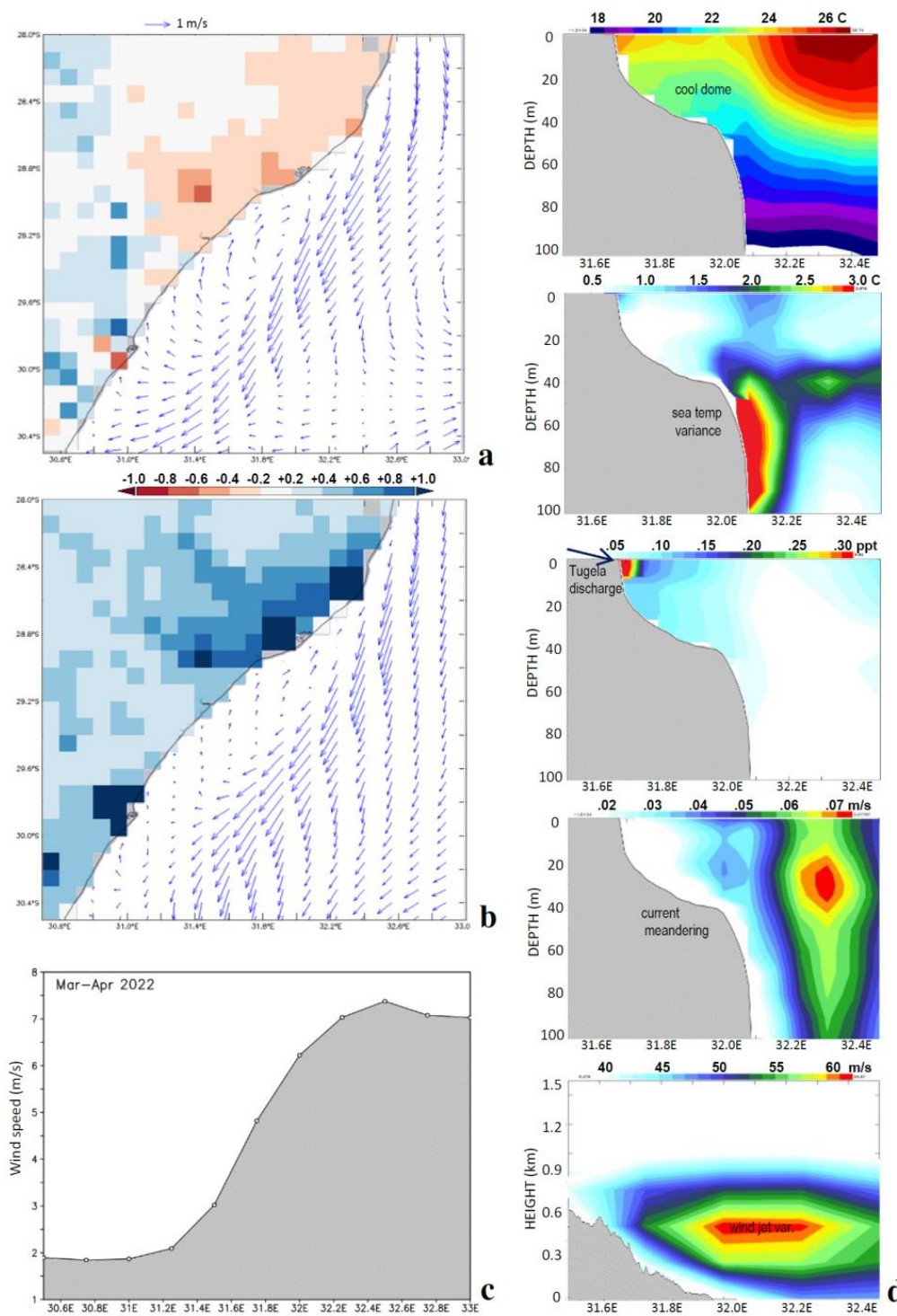

**Figure 9.** HYCOM currents in the mixed layer on: (**a**) 14 March and (**b**) 23 April 2022, and runoff anomalies (shaded mm/day). (**c**) Mean surface wind speed on 29.1°S over the 2-month period. (**d**) Right panels are vertical sections on 29.1°S of (top-down) sea temperature, and variance of: sea temp, salinity, U current (with depth), and V wind (with height); elevation profiles overlain.

Statistical context is provided by analysis of March–April 2022 means and variance in Figure 9c,d. Mean wind speeds along 29.1°S show a steep offshore increase from 2 m/s (31.2°E) to 7 m/s (32.2°E). Wind vorticity fluctuates from ~0 (summer) to $-10^{-5}$ s$^{-1}$ (winter, cyclonic, cf. Figure S1 lower) sustaining uplift over the Tugela Bank. Mean vertical sections on 29.1°S (Figure 9d) reflect a cool dome in the

20–40 m layer from 31.8 to 32°E underpinned by cyclonic current vorticity $-10^{-5}$ s$^{-1}$. Sea temperature variance >2 °C occurs on the shelf edge (32.1°E) in the 50–80 m layer according to HYCOM reanalysis. We infer that the cool dome is quite steady despite the alternating weather. Salinity variance is naturally high near the coast (>0.3 ppt, cf. Figure 9d) due to the Tugela discharge. Due to its buoyancy, the fresh plume penetrates only 10 m deep and is trapped within 20 km of the coast by longshore flow according to HYCOM reanalysis. Zonal current variance indicative of Agulhas meandering (>0.05 m/s) increases seaward of the shelf edge at 32°E, skirting and deforming the inshore gyre.

The lower panel of Figure 9d is an atmospheric section for March–April 2022. It shows meridional wind variance of 60 m$^2$/s$^2$ in the form of a low-level jet at 0.5 km elevation extending from 31.9 to 32.3°E. The longshore airflow is channeled to a width of ~40 km by steep terrain and atmospheric subsidence. The low-level jet is amplified by coastal lows and moist turbulence over the Agulhas Current.

### 3.4. Sediment Nutrification and Local Climate Trends

Supplementary Figure S2 illustrates a Hovmoller plot of red-band reflectance and river discharge in the MODIS era 2003–2012, highlighting the summer-runoff/winter-nutrification sequence, similar to Figure 2a. Maps for February and June 2011 (Figure S2) illustrate (again) how nutrients spread from the river mouth and shelf edge to a broad lens over the Tugela Bank.

The near-shore sediment budget depends on river discharge, coastal erosion, shelf bathymetry, and interaction of wind waves and littoral currents that induce seabed stress via turbulence and shear [64,65]. The product of sediment concentration and mixed layer current velocity yields transport rates of 0.3 kg m$^{-2}$/s for 0.3 m/s currents and 2.6 m wave heights, using the nomogram of [66], the lower panel Figure S2. The suspended sediment ~0.1 kg /m$^3$ is composed of fine sands, mud, and detritus with diameters <0.1 mm [16] derived from river discharge and reworking in the surf zone and adjacent shelf.

The turbulent lifting of near-shore sediment into the water column is quantified by seafloor shear stress during wind wave events. The study by [67] shows that $\rho(v'w')$ representing seawater density times the down-wave (~0.2 Hz) fluctuations of horizontal and vertical orbital velocity (or eddy covariance) exceed 1 N/m$^2$ under waves ~2.5 m height. The authors of [68] measured wave orbital velocities and show values of ~0.1 m/s at 30 m depth for wave lengths >100 m. In our case we have five spells of big waves with periods ~10 s: (date, Hs dir) 15 March, 4.1 m 191°; 30 March, 3.4 m 053°; 7 April, 3.7 m 203°; 12 April, 3.8 m 098°; 15 April, 3.7 m 201°; and 25 April 2022, 3.7 m 201°. Hence three weather events produce a northward flux and two weather events produce a westward flux. The study by [69] showed that wind wave turbulence penetrates deeper in isothermal than stratified conditions; big waves associated with cold air outbreaks are more effective at mobilizing sediments (cf. Figure 3a,f). In our case air temperatures decreased by 10 °C on 7 April and 25 April 2022. Turbid nutrification in the Natal Bight was induced by northward wave turbulence $\rho(v'w')$ accompanied by (latent heat flux) evaporation ~10 mm/day and MLD ~45 m.

Local climate changes are analyzed in Figure 10a,b by linear regression of field data from 1950 to 2021. Global warming trends over the 70 yr period are relatively weak in the Natal Bight: SST increases +0.015 °C/yr in contrast with +0.03 °C/yr over the interior highlands. ERA5 airflow trends exhibit an increase in northeasterly wind since 1950, resulting in more coastal upwelling that recirculates into the Natal Bight. However, runoff trends are downward especially in the Tugela Valley from January–March and suggest a reduction in river discharge and terrestrial nutrients. Historical records thus indicate the subtropical ridge is moving poleward and the Mascarene high is strengthening to the east of South Africa.

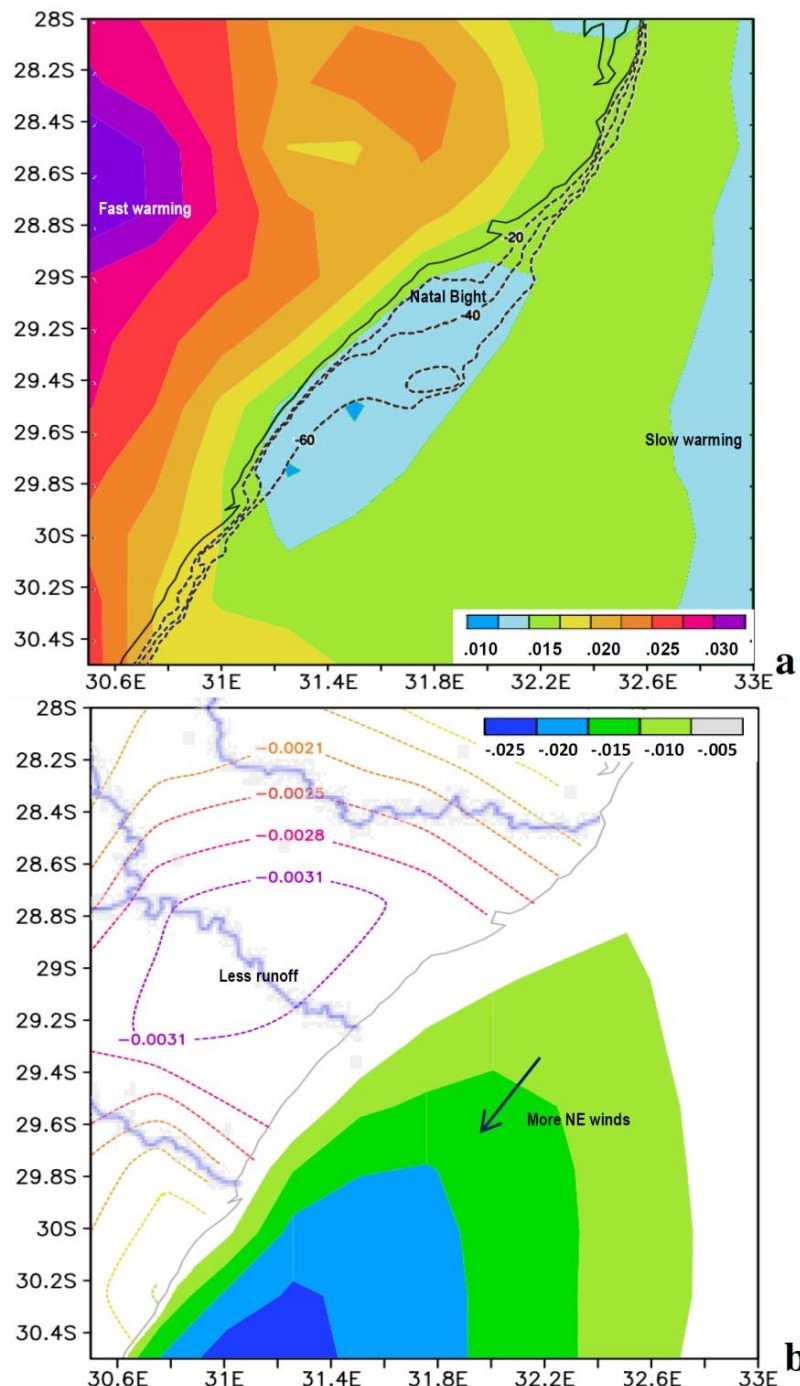

**Figure 10.** Local climate trends 1950–2021: (**a**) surface temperature (°C/yr) with shelf bathymetry, (**b**) runoff (contour mm/day yr$^{-1}$) and marine U V wind (shaded m/s yr$^{-1}$) based on linear regression of ERA5 field data.

## 4. Conclusions

The KwaZulu-Natal shelf edge is characterized by a narrow band of upwelling next to the warm Agulhas Current. Reversing longshore winds ~7 m/s and meandering poleward currents ~1 m/s pulse the system. Along the leeward coast that forms the Natal Bight, there is a shadow zone of weak cyclonic flow (cf. Figure 1c,d) that aggregates plankton, recycles nutrients, and sustains marine resources. A new outcome from this work is that red-band reflectance ~670 nm is significantly associated with the physical environmental and appears to be a useful proxy of productivity in the Natal Bight.



The surface heat balance reaches +70 W/m$^2$ in December, followed by river discharges ~3 M m$^3$/yr of fresh nutrient-rich water that peak in February. This induces a buoyant surface layer $\partial T/\partial z$ ~2 °C/35 m with MLD ~25 m that inhibits wind wave turbulence during summer. By contrast, winter (June–August) cooling Q ~ −95 W/m$^2$ and frequent cyclonic storminess deepen the mixed layer ~65 m and enable wind wave turbulence to scour the coastal seabed. The seasonal cycle is of high amplitude for many variables; runoff is constrained during winter by high coastal dunes, littoral drift, and infiltration. Summertime pulses of Tugela River discharge combine with salty coastal upwelling and winter recirculation to keep salinity in a narrow range ~35.37 ppt.

Our mean annual cycle analysis of daily longshore wind exceedances yielded an unexpected result: instead of northeasters in summer and southwesters in winter, we found amplification in spring (August–October, cf. Figure 3c) when troughs from the Atlantic Ocean and ridges from the Indian Ocean pulse the coastal weather and upwelling. Another key feature was cyclonic wind vorticity during winter (cf. Figure 3e) associated with land breezes that keep northeasterly winds outside the shelf edge (Figure S3). Daily wind vorticity shows negative spikes that relate to coastal lows advancing toward the Mascarene high (Figure S3). Our overall impression of seasonality is that sequential inputs have a knock-on effect across trophic levels that sustains year-round marine nutrification in the Natal Bight.

Intra-seasonal fluctuations are driven by meandering currents and changes in coastal weather accompanied by shelf waves. These were revealed in spectral analysis of Natal Bight daily winds, currents, and waves with significant ~5-day periods (cf. Figure 4a). Mixed layer currents follow meridional winds; northward air–sea momentum transfer in the Natal Bight is most efficient during spells of evaporation and big waves (cf. Figure 4c). The axis/shape/speed of the Agulhas Current plays a role in trapping and spinning the cyclonic gyre in the Natal Bight that is frequently disrupted by warm rings and anticyclonic filaments passing along the shelf edge, independent of weather-forcing. Such rapid changes make ship-based surveys a synoptic from a physical standpoint, but essential to understanding chemical nutrification and biological recycling.

New insights here derive from the use of daily 10 km resolution hindcast fields to understand air–land–sea fluxes in the shelf zone. Although recent studies [14] lament difficulties in hydrodynamic modeling and sparsity of data, advances in coupled modelling and satellite technology offer synoptic coverage of nearshore environmental conditions at unprecedented detail [70]. Operational in situ data collection that feeds global data assimilation would improve the reliability of CFS2, ERA5, HYCOM, etc., build confidence in local application, and guide the sustainable management of our marine resources.

**Supplementary Materials:** The following supporting information can be downloaded at: https://www.mdpi.com/article/10.3390/rs15051434/s1.

**Funding:** The author receives outcome-based support from the South African Department of Higher Education via the University of Zululand.

**Data Availability Statement:** An extensive spreadsheet is available from the author on request.

**Acknowledgments:** The IRI Climate Library, KNMI Climate Explorer, University Hawaii APDRC, and NASA Giovanni websites enabled data analysis. Tugela River discharge was updated from the SA Department of Water Affairs. Amos Mthembu from University Zululand provided valuable research material, and Ray Barlow of University Cape Town offered useful insights.

**Conflicts of Interest:** The author declares no conflict of interest.

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
