# Peer review of "Modulation of the Marine Environment in the Natal Bight"

_remotesensing, doi:10.3390/rs15051434_

Round 1

Reviewer 1 Report

This paper describes the modulation of the marine environment in the Natal bight.  The study area is located in a well known upwelling area but lacks details of the physical/biological processes.  This manuscript will add substantially to the knowledge of coastal upwelling in this important fisheries area.

The manuscript is very well written with good English composition.  The tables and figures are appropriate and support the text.  Several of the figures (e.g. Fig 3a and 8a) need better labeling of the vertical axes.

Author Response

all revisions are included in the feedback file

Reviewer 2 Report

In this study the author presents a detailed analysis of the Natal Bight and surrounding region. Whilst our knowledge of this region is far from complete, it is also not totally incomplete and the high-temporal resolution datasets used here build upon previous studies to provide further insight into the variability, and drivers of that variability, of the marine environment. Overall, the study is well written, appears scientifically sound and does add new knowledge on this part of South Africa’s coast. I have few comments.

Major comments:

Figures – The majority of figures need to be remade to correct for omissions (e.g. axis labels, colour scale labels, font size etc). Many are not acceptable in their current form, can contain illegible text or look to have been cut and pasted from elsewhere.

Figure 1 – Suggest reorganising figure into a 3 (wide) by 2 (high) arrangement to better fit page and allow minor increase in subplot size. Many labels appear missing or cut off in individual subplots. Figure 1d is missing all place name labels. Add shelf edge / 200m bathymetric contour to all plots. Legend in 1a is illegible, colour bar label in 1b cut off.

Figure 2 – Subplot 2a is limited to 2012-2018, despite what text states, axis labels on panel b are unreadable, colour map made it very difficult to read and colour scale missing. Also based on description in text, and implication of Fig 2b the lower right photo appears to show the river plume curving to the left. This may be the perspective of the photo but appears contrary to the expectation that the plume would move southwards (i.e to the right from the observers perspective). Is this correct?

Figure 3 – 3a y-axis labels need correcting, 3f contains unreadable text in lower left corner,  3i & 3j have same subplot label, and lack x-axis labels.

Figure 4 – Subplot 4d and 4e not correctly labelled (See text P10 Para1 which refers to Fig 4e)

Figure 5 – Figure 5b yaxis labels, xaxis label missing (‘Year’)

Figure 7 – Axis labels are small and difficult to read

Figure 8 appears incomplete with missing subplots, floating subplot labels, floating axis or missing axis labels. Please redo.

Figure 9 – 9a Truncated colour bar labels, subplot labels for 9c and 9d appear in wrong place.

Figure S1 – Axis labels unreadable, incomplete or missing.  

Section 3.2 & P19 – The observation that red-band fluorescence is a better proxy for productivity requires clarification and expansion but contrary to the inferred relationship to productivity most likely suggests that the observed signal is related to sediment loading (as per P3 para 5) rather than productivity.

Nutrient data – Whilst I will accept that this study is first and foremost based upon remote sensing or modelled outputs, there is ample reference throughout the text to nutrification and/or nutrient levels, but no actual evidence to support the arguments made. I would be more confident if some supporting observations of nutrient concentrations could be included from the literature to help clarify the presented interpretation. For e.g. on P12 reference is made to ‘Tugela Valley runoff leading nutrients’. Are there any data from the river itself to support this?

Minor comments:

(With apologies, the lack of line numbers makes the following difficult to precisely locate)

Page 2 3rd paragraph – This is entirely pedantic but the word unique is a qualified absolute. Something cannot be ‘rather unique’. It either is or is not unique.

P5 3rd para – Missing word? “Our record length of 240…….”

P6 4th para (section 3.1) – The statement that the mixed layer is >40 m implies this is true everywhere, whereas Fig 1b shows only a localised area where this is true. Please correct to include estimate of variability and/or range in mixed layer depths.

P6 5th para – The text states that fig 2a,b covers the VIIRS era (2012-2022), or that there are low values of red-band reflectance in 2018-2021. Figure 2 however only covers the period 2012-2018. Please correct.

P9 Para2 – missing text (~0.8 10-5)

P12 Para 1 – Text refers to Fig 6d, Fig 6 only shows panels a-c.

P17 Para 3 – If the VIIRS (Fig 2) and MODIS data (Fig S2) are comparable what is the value of Fig S2? (other than to extend the time series). Appears unnecessary.

P19 1st line – Typo environment ?

P19 Para 2 – salinity should be dimensionless (under EOS 80) or have units of g/kg (under TEOS 10). Regardless, units of ppt are redundant.

Author Response

(The authors gave the same response as above.)

Reviewer 3 Report

The proposed study investigates the dynamics of the coastal strip of the "Natal Bight". The study is conducted with high scientific rigor, supported by state-of-art material and datasets. The scientific analysis and conclusions are significant and characteristic for the study area. The work has a high scientific relevance in the field of marine remote sensing and oceanographic dynamics.

1) I suggest to the author to add an image that makes it easier to locate the bay, general circulation at mesoscale, atmospheric and marine, topography and bathymetry.

2) improve the quality of the figures. A "mask" with some letters and numbers is observed "in the interior of the coast".

3) Figure 1 f) rivers are "in the sea". Coordinates and colors are moved or shifted with respect to the panels. Correct the image.

4) the WavwWatch3 model is usually indicated with the acronym WW3.

5) Figure 5,8 : Improve image quality. Vectors outside the plot etc.

6) Figure 6: longitudes overwritten incorrectly

7) Figure 7: very small fonts.

8) Figure 9: correct the position of the colorar.

Author Response

(The authors gave the same response as above.)
